

# The below-ground carbon and nitrogen cycling patterns of different mycorrhizal forests on the eastern Qinghai-Tibetan Plateau

Miaomiao Zhang[1,2], Shun Liu[1,2], Miao Chen[1,2], Jian Chen[1,2], Xiangwen Cao[1,2], Gexi Xu[1,2], Hongshuang Xing[1,2], Feifan Li[1,2] and Zuomin Shi[1,2,3,4]

[1] Key Laboratory of Forest Ecology and Environment of National Forestry and Grassland Administration, Ecology and Nature Conservation Institute, Chinese Academy of Forestry, Beijing, China

[2] Miyaluo Research Station of Alpine Forest Ecosystem, Lixian County, Sichuan, China

[3] Co-Innovation Center for Sustainable Forestry in Southern China, Nanjing Forestry University, Nanjing, China

[4] Institute for Sustainable Plant Protection, National Research Council of Italy, Torino, Italy

Corresponding author
Zuomin Shi, shizm@caf.ac.cn

## ABSTRACT

Mycorrhizal fungi can form symbiotic associations with tree species, which not only play an important role in plant survival and growth, but also in soil carbon (C) and nitrogen (N) cycling. However, the understanding of differences in soil C and N cycling patterns among forests with different mycorrhizal types is still incomplete. In order to determine the similarities and differences of soil C and N cycling patterns in different mycorrhizal forest types, three primary forests dominated by ectomycorrhizal (EcM), arbuscular mycorrhizal (AM) and ericoid mycorrhizal (ErM) trees respectively were studied on the eastern Qinghai-Tibetan Plateau. Indicators associated with soil C and N cycling, including leaf litter quality, soil C and N contents, soil C and N fluxes, and soil microbial biomass C and N contents were measured in each mycorrhizal type forest. The results showed that leaf litter quality was significantly lower with high C:N ratio and lignin: N ratio in ErM forest than that in AM and EcM forests. Soil $CO_2$ flux (508.25 ± 65.51 mg m$^{-2}$ h$^{-1}$) in AM forest was significantly higher than that in EcM forest (387.18 ± 56.19 mg m$^{-2}$ h$^{-1}$) and ErM forest (177.87 ± 58.40 mg m$^{-2}$ h$^{-1}$). Furthermore, soil inorganic N content was higher in the AM forest than that in EcM and ErM forests. Soil net N mineralization rate (−0.02 ± 0.03 mg kg$^{-1}$ d$^{-1}$) was lower in ErM forest than that in EcM and AM forests. We speculated that AM and EcM forests were relatively characterized by rapid soil C cycling comparing to ErM forest. The soil N cycling in EcM and ErM forests were lower, implying they were 'organic' N nutrition patterns, and the pattern in ErM forest was more obvious.

## INTRODUCTION

Most angiosperms are associated with mycorrhizal fungi, which can increase their access to nutrients and water or enhance the stress resistant ability (*Cheeke et al., 2017*; *Tedersoo & Bahram, 2019*). Mycorrhizal fungi are generally divided into three major types according to

their anatomy and function, such as ectomycorrhizal (EcM), arbuscular mycorrhizal (AM) and ericoid mycorrhizal (ErM) fungi (*Read & Perez-Moreno, 2003*). Increasing evidence indicates that functional variations of EcM, AM and ErM fungi may drive soil C and nutrient cycling in varying degrees (*Phillips, Brzostek & Midgley, 2013*; *Tedersoo, Bahram & Zobel, 2020*).

The soils in EcM- and ErM-dominated ecosystems have higher C contents than in AM-dominated ecosystems due to the specific N uptake preference of EcM and ErM fungi (*Averill & Finzi, 2011*; *Cavagnaro, Barrios-Masias & Jackson, 2012*; *Adamczyk et al., 2016*). The effect of mycorrhizal types on soil C contents may be regulated by the chemical differences of plant litter associated with EcM, AM and ErM. Compared to AM plants, EcM and ErM plants have relatively lower growth rates as well as lower leaf litter quality, *i.e.*, higher C:N or lignin:N (*Taylor, Lankau & Wurzburger, 2016*; *Read, Leake & Perez-Moreno, 2004*). Therefore, the leaf litter decomposition rates of EcM and ErM plants are relatively slower than that of AM plants, resulting in increase of soil C content in EcM- and ErM-dominated ecosystems (*Lin et al., 2016*; *Wurzburger & Hendrick, 2009*). Furthermore, *Clemmensen et al. (2021)* indicated that ErM plants decomposed slower than EcM plants and tended to accumulate more soil organic matter. On the contrary, some studies declared that AM soil had more C and N in temperate broadleaved forest (*Craig et al., 2018*). That is possibly attributed to enhancing the mineral-associated organic matter (MAOM) by promoting production and stabilization of microbial residues in AM forest (*Rillig, 2004*; *Cotrufo et al., 2013*). For soil C fluxes, the significant differences in different mycorrhizal forests may be caused by the result of the interactions between mycorrhizal associations and microorganisms (*Shi, Wang & Liu, 2012*; *Hughes et al., 2008*; *Heinemeyer et al., 2007*). Mycorrhizal exudates could promote soil organic matter decomposition and thus increase soil $CO_2$ flux ('Rhizosphere priming hypothesis', *Brzostek et al., 2015*). Study also shows that mycorrhizal fungi could compete directly with free-living decomposers for organic N which decreasing soil $CO_2$ flux ('Gadgil effect', *Gadgil & Gadgil, 1971*). Furthermore, niche assembly indicates that mycorrhizal fungi have different ecological niches from other microorganisms and there are no obvious interactions between mycorrhizal fungi and other microorganisms, which do not affect soil $CO_2$ flux (*Lindahl et al., 2007*).

Compared to the complex differences of soil C cycling, the differences of soil N cycling among different mycorrhizal forests are systematic (*Wang, Wang & Zhang, 2017*). In general, AM fungi adapt to ecosystems with high inorganic N content and lack proteolytic ability, while EcM and ErM fungi adapt to ecosystems with low N cycling rate and rich organic matter (*Hobbie & Högberg, 2012*). Similarly, *Phillips, Brzostek & Midgley (2013)* proposed the mycorrhizal-associated nutrient economy (MANE) framework that AM forest associated with the inorganic N economy and EcM forest with organic N economy. The unique effects of different mycorrhizal trees on soil N cycling are attributed mainly to the changes of their related mycorrhizal fungi traits (*Read & Perez-Moreno, 2003*; *Averill, 2016*). Previous studies showed that EcM and ErM fungi could exudate extracellular enzymes to degrade recalcitrant organic nutrients that could not be absorbed by plants (*Orwin et al., 2011*; *Ward et al., 2021*) and ErM fungi had stronger degradation ability than EcM fungi (*Read & Perez-Moreno, 2003*). AM fungi seem to lack the ability to express C and

N degrading enzymes comparing to EcM and ErM fungi (*Tisserant et al., 2013*). However, AM fungi could increase the absorption of dissolved and inorganic nutrients by changing the traits (biomass and growth rate) of host plant root system (*Bennett & Groten, 2022*).

Different mycorrhizal fungi have great differences in species diversity, C demands (*Orwin et al., 2011*), and the ability of enzymes to access different forms of N (*Hobbie & Högberg, 2012*). Previous studies on the differences of soil C and N cycling patterns in different mycorrhizal ecosystems focused predominantly on the global scale (*Tedersoo & Bahram, 2019*; *Lin et al., 2016*), and a few studies were carried out on a small scale (*Cheeke et al., 2017*; *Taylor, Lankau & Wurzburger, 2016*). Additionally, most of these investigations focused mainly on EcM- and AM-dominated ecosystems (*Keller & Phillips, 2018*; *Fang et al., 2020*). Comparative study on soil C and N cycling patterns among EcM-, AM- and ErM-dominanted ecosystems is still incomplete.

The eastern Qinghai-Tibetan Plateau presents a unique natural environment which plays an important role in soil and water conservation, climate regulation and biodiversity protection (*Wang et al., 2007*). It is an important ecological barrier in the middle and upper reaches of the Yangtze River (*Chen, 2019*). In-depth explorations of the biogeochemical cycle of the forest ecosystems with different mycorrhizal types are essential to understand their ecological functions. To understand more the below-ground C and N cycling patterns of different mycorrhizal forests on the eastern Qinghai-Tibetan Plateau, key parameters related to soil C and N cycling (*Lin et al., 2016*) of three primary forests (including EcM, AM and ErM) were studied. Our hypotheses were: (1) AM forest had faster C cycling than EcM and ErM forests ; (2) AM forest had faster N cycling (soil C:N, soil net nitrification rate and soil net N mineralization rate) dominated by soil inorganic N ($NH_4^+$-N and $NO_3^-$-N) forms, EcM and ErM forests relied more on soil organic N.

## MATERIALS & METHODS

### Study site

The study was conducted in the upper reaches of the Minjiang River, western Sichuan Province. This area is located in the outermost part of the fold belt on the eastern Qinghai-Tibetan Plateau (*Xu et al., 2021*). The average annual temperature is 2∼4 °C, the highest temperature in summer is 23.7 °C, and the lowest temperature in winter is −18.1 °C. Annual precipitation is 700∼1,000 mm and concentrated mainly in the growing season (*Chen, 2019*). The soil in this area is defined as mountain brown soil, mountain brown cinnamon soil and subalpine meadow soil according to the Chinese soil taxonomic classification (*Liu, 2010*; *Feng et al., 2017*; *Chen, 2019*).

### Experimental design

Three primary forests with different mycorrhizal types under similar soil and climate conditions, including *Abies faxoniana* primary forest (EcM), *Cupressus chengiana* primary forest (AM) and *Rhododendron phaeochrysum* primary forest (ErM) were selected in June 2018. Eight 15 m × 15 m sample plots (≥90% the dominant tree species by basal area in each sample plot) for each forest type were randomly set. In each forest type, the distance

Table 1 The basic stand information of the three forests. Data in the table were mean ± standard error ($n = 8$ in each case).

| Forest type | Abies faxoniana primary forest | Cupressus chengiana primary forest | Rhododendron phaeochrysum primary forest |
|---|---|---|---|
| Mycorrhizal type | EcM | AM | ErM |
| Geographical coordinate | 31°35′N, 102°48′E | 31°54′N, 102°2′E | 31°53′N, 102°46′E |
| Elevation (m) | 3,295 | 2,802 | 3,805 |
| Stand DBH (cm) | 34.87 ± 13.56 | 21.63 ± 1.88 | 4.36 ± 0.50 |
| Stand height (m) | 18.52 ± 1.14 | 8.16 ± 0.79 | 2.64 ± 0.16 |
| Canopy density | 0.90 | 0.90 | 0.95 |
| Soil pH | 5.48 | 6.70 | 5.01 |

between any two sample plots was more than 50 m. The basic stand information of each forest type was given in Table 1.

## Soil and leaf litter sampling

Soil (0–10 cm) samples were collected from four corners and central position of each plot with a soil drill in July 2019. The five soil samples were mixed into a zipper storage bag and transported to laboratory in ice box within 3 h. Moreover, Five to six trees without diseases and pests in each plot were selected randomly. Leaf litter without decay or degradation under each tree was sampled (over 100 g) and mixed together to form one sample per plot (*Barajas-Guzmán & Alvarez-Sánchez, 2003*).

## Soil N mineralization

Five polyvinyl chloride cores (PVC cores, 15 cm height and 5 cm diameter) were buried into depth of 10 cm in the vicinity of the soil sampling location in July 2019 (*Becker et al., 2015*). Top of the PVC core was covered with permeable plastic film and bottom was with gauze to segregate water and allowed gas flow (*Kong et al., 2019*). Soil samples in the PVC cores were taken out after *in situ* for one month and sent to the laboratory for measurements. Soil net N mineralization rate and soil net nitrification rate were quantified by the difference between $NH_4^+$-N and $NO_3^-$-N per month (*Wang et al., 2017*).

## Soil respiration

Soil respiration was measured by static chambers and the gas chromatography technique. In September 2018, three static chamber bottoms were installed in each sample plot. The polyvinyl chloride cores (PVC cores, 10 cm height and 25 cm diameter) were buried 5 cm underground and reduced soil compaction in this process. The PVC cores were hollow and their tops and bottoms were unwrapped (*Heinemeyer et al., 2007*; *Tomè et al., 2016*). The living plants in the polyvinyl chloride cores were removed to reduce the impact of surface vegetation on soil $CO_2$ flux (*Wang et al., 2010*). In July 2019, a PVC portable opaque chamber (cylindrical, 30 cm height and 25 cm diameter) fitted with a fan to mix the air was attached to bottom of static chamber (*Wu et al., 2018*). For gas collection, a rubber tube (20 cm length and 5 mm diameter) was attached to top of the portable opaque chamber, which was kept closed throughout sampling to ensure airtightness (*Wanyama et al., 2019*). Gas samples were taken on sunny days between 9 am and 11 am (*Cheng et al., 2010*). A

gas sample was extracted from the static chamber using a gas-tight syringe (100 ml) at 0 min, 15 min, 30 min, and 45 min, respectively and injected into a gas preservation bag. The gas samples were sealed and transported to the laboratory within 48 h (*Wang et al., 2010*). Carbon dioxide concentration was analyzed by a gas chromatograph equipped with a flame ionization detector (Agilent4890D; Agilent Technologies, Santa Clara, CA, USA). The calculation method of $CO_2$ fluxes was referred to *Chen et al. (2021)*.

## Measurements of soil and leaf litter

Soil samples were divided into two parts after passing through a 2-mm sieve; one was stored at $-20\ °C$ to measure the dissolved organic C (DOC), $NH_4^+$-N, $NO_3^-$-N, microbial biomass C (MBC) and microbial biomass N (MBN). Microbial biomass C:N ratio could be used to characterize the ratio of soil fungi to bacteria (*Bardgett et al., 2005*). The other part of soil was dried naturally at room temperature to measure soil organic C (SOC), soil total N (TN) and soil pH. In addition, leaf litter samples were dried at 70 °C to obtain constant weight, then ground and stored for analyzing organic C, TN and lignin contents.

Soil DOC content was measured by TOC analyzer (TOC-5000 Analyzer, Shimadzu and Kyoto, Japan) setting the 1:5 ratio of soil material to deionized water. The $NO_3^-$-N and $NH_4^+$-N were determined by automatic flow injection analyzer (FIAstar 5000 Analyzer, FOSS, Hillerød, Denmark). The MBC and MBN were determined by chloroform fumigation -$K_2SO_4$ extraction. SOC and leaf litter organic C contents were measured using the wet oxidation method with $K_2Cr_2O_7$ and $H_2SO_4$, and $FeSO_4$ titration. TN content was determined using the Kjeldahl method (*Liu et al., 2021*). Soil pH was determined by the glass electrode meter method setting the 1:2.5 (w/v) ratio of soil material to deionized water. Leaf litter lignin content was analyzed using concentrated sulfuric acid method (*Singh et al., 2021*). The C:N ratio was the ratio of organic C to TN.

## Statistical analysis

A one-way analysis of variance (ANOVA) was performed to examine the differences of litter quality, contents of soil C and N, soil C and N fluxes and microbial biomass C and N among the three forests. Tukey's HSD method was used for multiple comparisons. SPSS was used for data processing and analysis, and Origin 8.0 was used to create figure.

# RESULTS

## Leaf litter quality

Leaf litter in ErM forest had significantly higher level in lignin: N ratio ($45.02 \pm 5.59$) than that in AM forest ($33.08 \pm 2.13$) and EcM forest ($26.63 \pm 5.29$) ($P < 0.05$) (Fig. 1A). Similarly, leaf litter C:N ratio in ErM forest was the highest ($83.05 \pm 10.98$), followed by AM forest ($60.19 \pm 5.81$) and EcM forest ($44.97 \pm 9.64$) ($P < 0.05$) (Fig. 1B).

## Contents of soil C and N

There were no significant differences in SOC contents among the three forests (Fig. 2A). Soil DOC content in EcM forest ($106.17 \pm 16.02$ mg $kg^{-1}$) was significantly higher than that in AM forest ($79.89 \pm 20.98$ mg $kg^{-1}$) and ErM forest ($70.15 \pm 5.61$ mg $kg^{-1}$) ($P <$
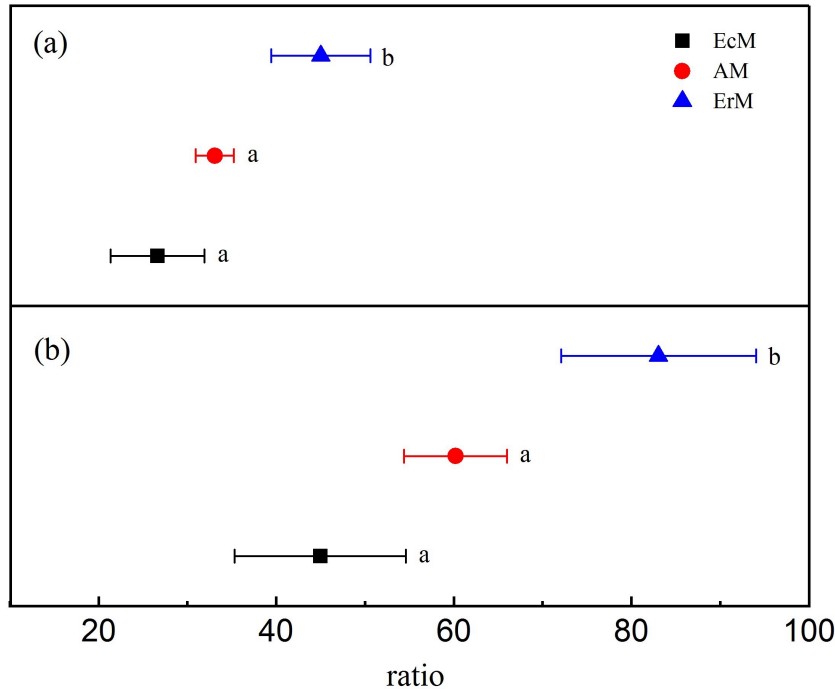

**Figure 1 Differences in leaf litter quality among the three forests.** Leaf litter lignin: N ratios (A) and leaf litter C:N ratios (B) in *Abies faxoniana* primary forest (EcM), *Cupressus chengiana* primary forest (AM) and *Rhododendron phaeochrysum* primary forest (ErM). Data in the figure were mean ± standard errors (parallel bars; $n = 8$ in each case). a, b and c indicated significant differences among the three forests ($P < 0.05$) according to a one-way analysis of variance.

0.05) (Fig. 2D). Moreover, the soil C:N ratio was also significantly higher in EcM forest than that in AM and ErM forests ($P < 0.05$) (Fig. 2C).

Soil TN content in EcM forest was significantly lower than in other forests ($P < 0.05$) (Fig. 2B). Furthermore, soil $NH_4^+$-N content in EcM forest ($4.74 \pm 2.40$ mg kg$^{-1}$) was significantly lower than that in AM forest ($10.26 \pm 2.79$ mg kg$^{-1}$) and ErM forest ($12.61 \pm 2.57$ mg kg$^{-1}$) ($P < 0.05$) (Fig. 2E). AM forest had the highest soil $NO_3^-$-N content and there was no significant difference between EcM and ErM forests (Fig. 2F).

## Soil C and N fluxes

There were significant differences of soil $CO_2$ flux among AM forest ($508.25 \pm 65.51$ mg m$^{-2}$ h$^{-1}$), EcM forest ($387.18 \pm 56.19$ mg m$^{-2}$ h$^{-1}$) and ErM forest ($177.87 \pm 58.40$ mg m$^{-2}$ h$^{-1}$) ($P < 0.05$) (Fig. 3A). Moreover, there were no significant differences in soil net nitrification rates (NN) among the three forests (Fig. 3B). Soil net N mineralization rate (NNM) in ErM forest was negative ($-0.02 \pm 0.03$ mg kg$^{-1}$ d$^{-1}$) and had significant differences with the other two forests ($P < 0.05$) (Fig. 3C).

## Soil microbial biomass C and N

The MBC and MBN contents in ErM forest were significantly higher than those in AM forest ($P < 0.05$) but there were no significant differences with EcM forest (Figs. 4A and

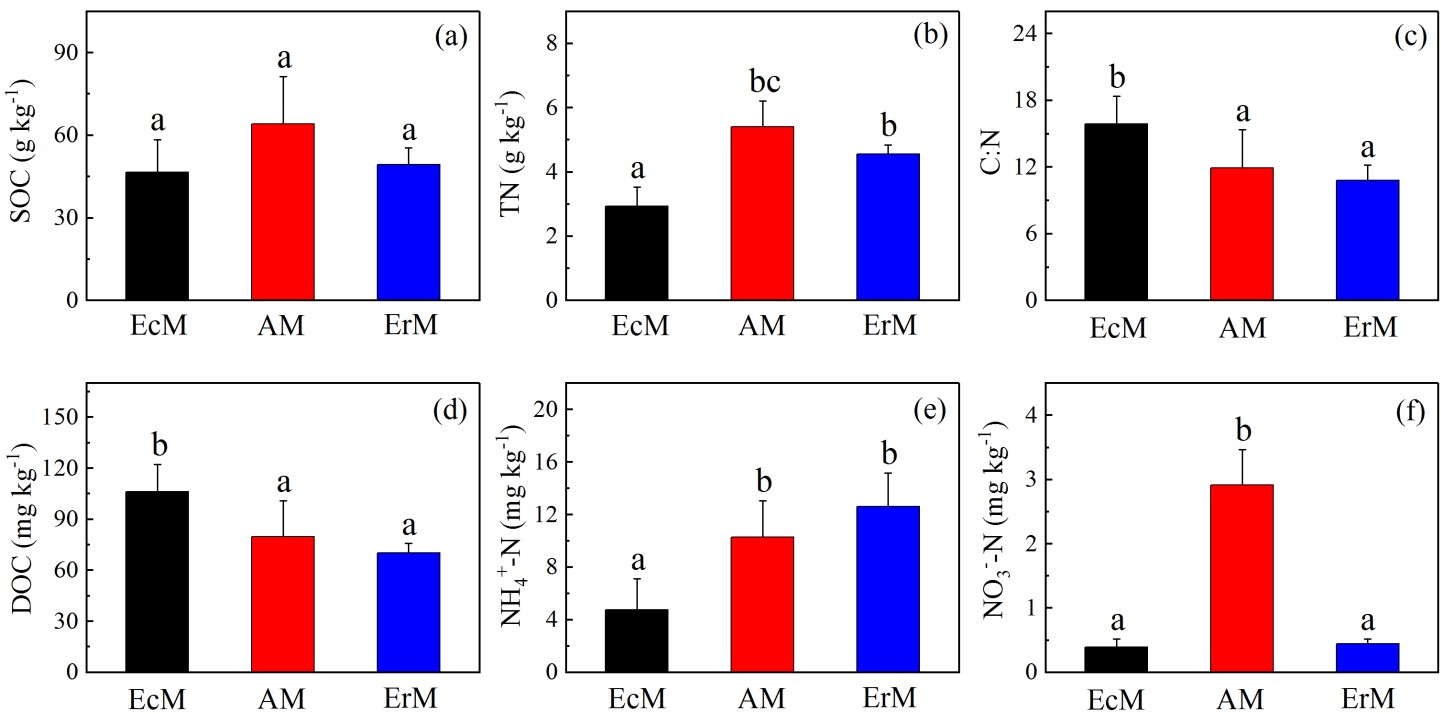

**Figure 2** **Differences in contents of soil C and N among the three forests.** SOC contents (A), soil TN contents (B), soil C:N ratios (C), mean soil DOC contents (D), soil $NH_4^+$-N contents (E) and soil $NO_3^-$-N contents (F) in *Abies faxoniana* primary forest (EcM), *Cupressus chengiana* primary forest (AM) and *Rhododendron phaeochrysum* primary forest (ErM). Data in the figure were mean ± standard errors (vertical bars; $n = 8$ in each case). a, b and c indicated significant differences among the three forests ($P < 0.05$) according to a one-way analysis of variance.

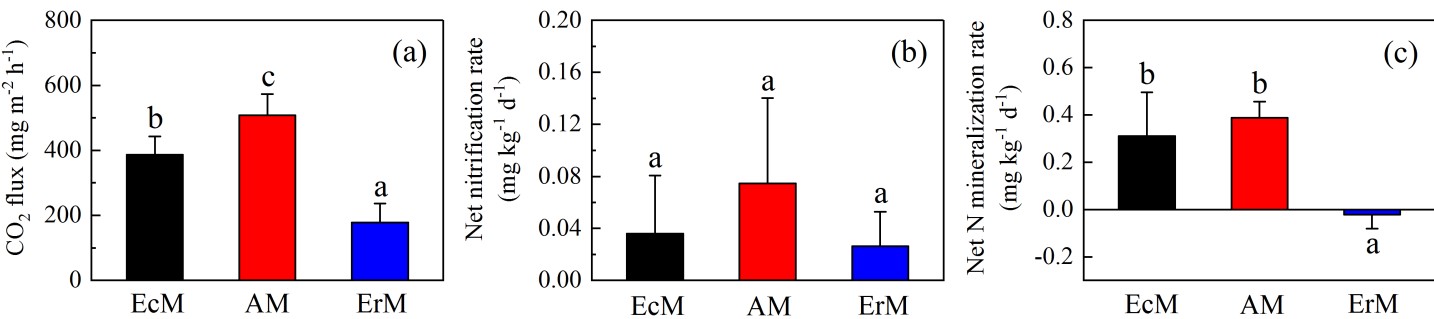

**Figure 3** **Differences in soil C and N fluxes among the three forests.** Soil $CO_2$ fluxes (A), soil net nitrification rates (B) and soil net N mineralization rates (C) in *Abies faxoniana* primary forest (EcM), *Cupressus chengiana* primary forest (AM) and *Rhododendron phaeochrysum* primary forest (ErM). Data in the figure were mean ± standard errors (vertical bars; $n = 8$ in each case). a, b and c indicated significant differences among the three forests ($P < 0.05$) according to a one-way analysis of variance.

4B). There was significant difference in MBC:MBN ratio between ErM forest (3.36 ± 1.03) and AM forest (2.73 ± 0.75) ($P < 0.05$), while the ratio in EcM forest (3.08 ± 0.91) had no significant difference with the other two forests (Fig. 4C).

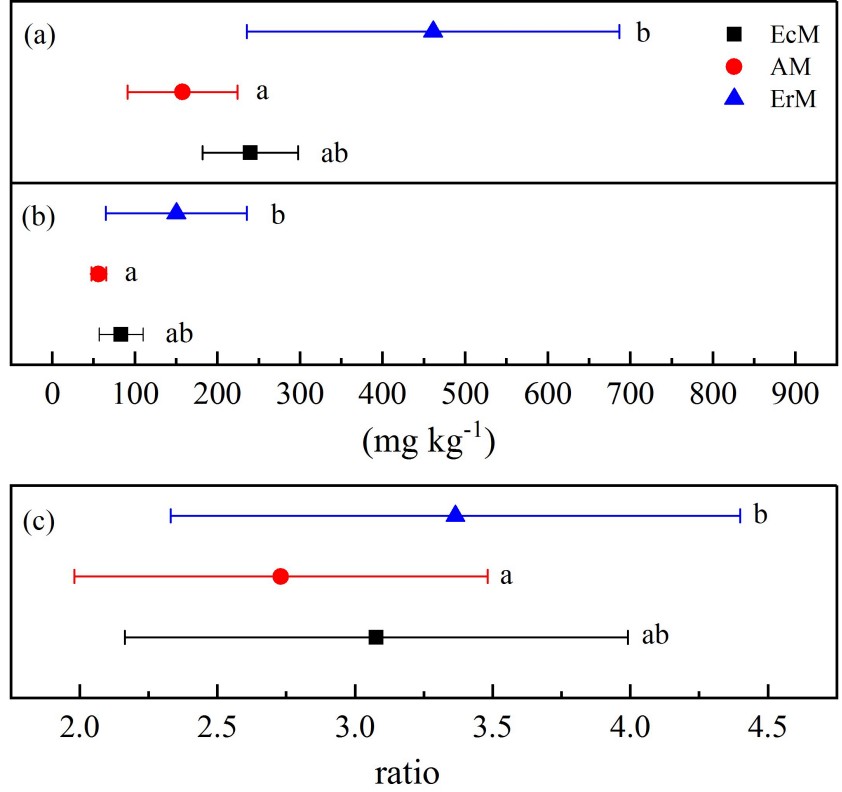

**Figure 4** **Differences in soil microbial biomass C and N contents among the three forests.** Soil microbial biomass C contents (A), soil microbial biomass N contents (B) and soil microbial biomass C:N ratios (C) in *Abies faxoniana* primary forest (EcM), *Cupressus chengiana* primary forest (AM) and *Rhododendron phaeochrysum* primary forest (ErM). Data in the figure were mean ± standard errors (parallel bars; $n = 8$ in each case). a, b and c indicated significant differences among the three forests ($P < 0.05$) according to a one-way analysis of variance.

## DISCUSSION

### Soil C cycling in different mycorrhizal forests

Compared to EcM and ErM forests, the AM forest had the highest soil $CO_2$ flux (Fig. 3A). Previous studies showed that EcM and ErM fungi preferred and decomposed selectively soil organic N (*Lin et al., 2016*; *Ward et al., 2021*). This nutrient strategy would drive N limitation of soil free-living decomposers, thereby slowing down the soil respiration (*Averill, Turner & Finzi, 2014*). Meanwhile, AM forest had lower microbial biomass C:N ratio than ErM forest in our study (Fig. 4C). It implied that the ratio of soil fungi to bacteria in AM forest was lower than that in ErM forest. AM forest tend to have bacteria-dominated food webs, which may result in AM forest having greater heterotrophic respiration (*Taylor, Lankau & Wurzburger, 2016*).

Simultaneously, the ErM forest had the slowest soil $CO_2$ flux in our study, which might be related to the quality of its leaf litter (Figs. 1 and 3A). Comparing to AM and EcM forests, leaf litter in ErM forest had higher C:N ratio and lignin: N ratio (Fig. 1) which could represent lower quality of leaf litter (*Midgley, Brzostek & Phillips, 2015*). The lower

quality may lead to accumulation of plant-derived compounds in the soil, which decreases soil $CO_2$ flux (*Taylor, Lankau & Wurzburger, 2016*; *Keller & Phillips, 2018*). In addition, as one of the main pathways of soil C output, soil DOC is mainly related to soil characteristic, decomposition of litter and humus (*Kindler et al., 2011*). Our results showed that soil DOC content in EcM forest was higher (Fig. 2D). We speculated that higher quality of leaf litter might be one of the reasons for the higher soil DOC content in EcM forest (Figs. 1 and 2D). Overall, our results showed that soil C cycling in AM and EcM forests were relatively faster than that in ErM forest (Fig. 5).

## Soil N cycling in different mycorrhizal forests

Our results showed that soil N cycling was more open in the AM forest than that in EcM and ErM forests (Fig. 5). These were reflected mainly in soil C:N, soil inorganic N content, soil net nitrification rate and soil net N mineralization rate (Figs. 2C, 2E, 2F, 3B and 3C). The more open N cycle pattern of AM forest confirms the long-held view that EcM, AM and ErM plants and related fungi differ in their acquisition, utilization and impact on soil N (*Lin et al., 2016*; *Craig et al., 2018*). In our study, soil C:N was lower in AM forest than that in EcM forest (Fig. 2C). Previous researches indicated that EcM fungi could obtain stable C sources from hosts (*Smith & Read, 2008*; *Lindahl & Tunlid, 2015*). As a result, EcM fungi could use selectively organic substrates to obtain N which might lead to higher soil C:N and slower N cycling (*Lin et al., 2016*). Our results showed that the AM forest had higher soil inorganic N content (Figs. 2E and 2F) which was consistent with previous study (*Tedersoo & Bahram, 2019*). Additionally, there were no significant differences in soil net nitrification rates among AM, EcM and ErM forests (Fig. 3B). Soil net N mineralization rate was lower in the ErM forest and there was no significant difference between AM and EcM forests (Fig. 3C). These results were not entirely consistent with previous studies, because the examinations pointed that soil net nitrification rate and soil net N mineralization rate in AM forests were higher than those in EcM forests (*Saifuddin et al., 2021*; *Lin et al., 2016*). However, there were other studies showed that EcM trees had stronger positive effect on soil net nitrification rate and soil net N mineralization rate than AM trees (*Phillips & Fahey, 2006*). *Chen et al. (2018)* indicated that there might be variations in soil net nitrification rate and soil net N mineralization rate in AM and EcM forests with different dominant tree species.

Previous studies showed that AM fungi absorbed primarily soil inorganic N (*Read & Perez-Moreno, 2003*; *Tedersoo, Bahram & Zobel, 2020*). Meanwhile, some studies suggested that EcM and ErM fungi had genetic potential to produce oxidase and glycoside hydrolase which could decompose recalcitrant organic matter (*Op De Beeck et al., 2018*; *Ward et al., 2021*) and mobilized N from organic matter (*Orwin et al., 2011*; *Shah et al., 2016*). *Phillips, Brzostek & Midgley (2013)* proposed that there was inorganic N economy in AM-dominated ecosystem but organic N economy in EcM-dominated ecosystem. In our study, more open soil N cycling implied that there was inorganic N pattern in AM forest (Fig. 5). In addition, our results showed that soil net N nitrification rate in ErM forest was positive and soil net N mineralization rate was negative (Figs. 3B and 3C), indicating that soil net ammonification rate in ErM forest was negative. Consumption of soil $NH_4^+$-N

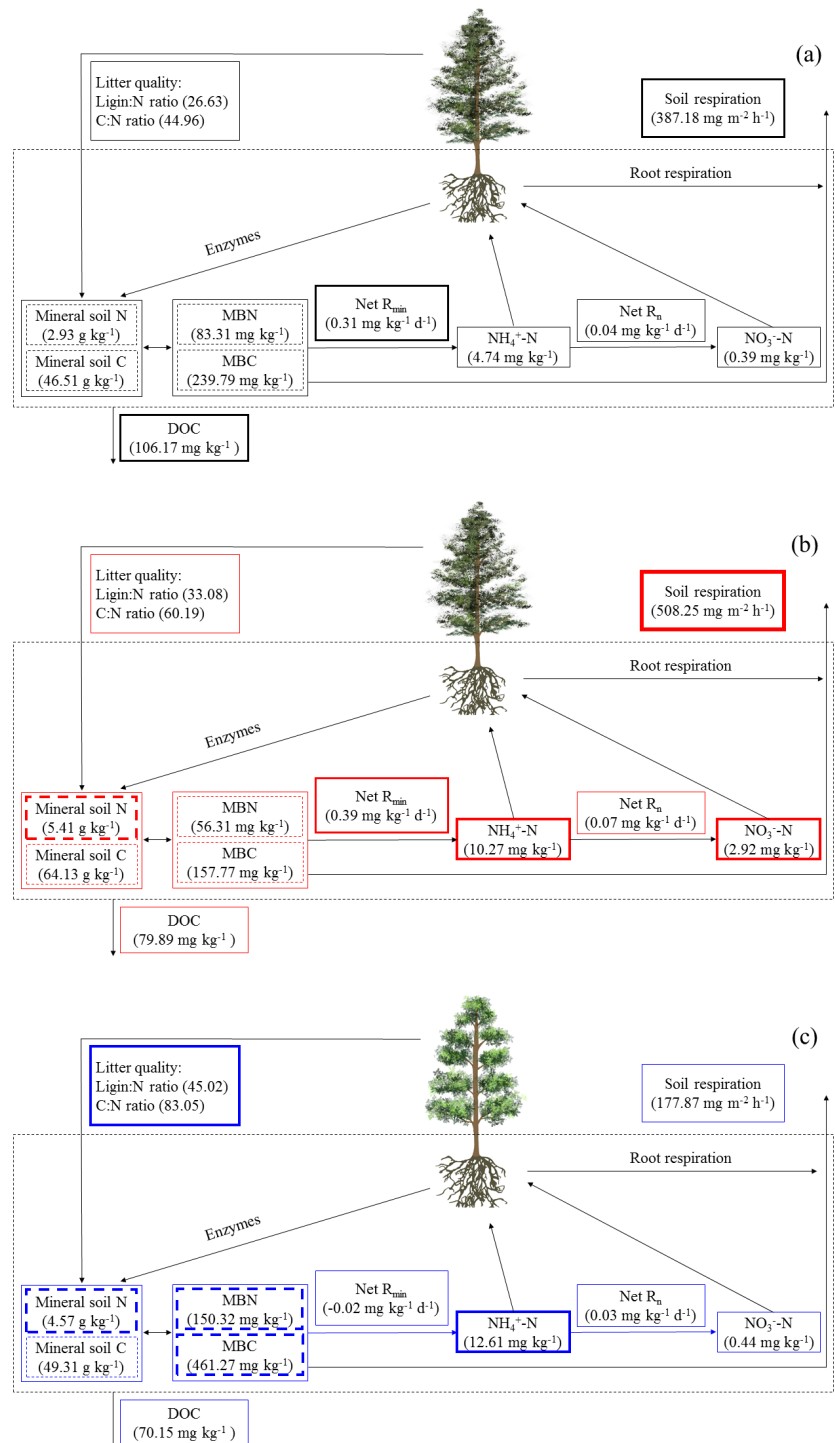

**Figure 5** **The soil C and N cycling in *Abies faxoniana* primary forest (EcM) (A), *Cupressus chengiana* primary forest (AM) (B) and *Rhododendron phaeochrysum* primary forest (ErM) (C).** Data in the figure were mean values ($n = 8$ in each case). DOC, dissolved organic C; MBC, microbial biomass C; MBN, microbial biomass N; Net $R_{min}$, net N mineralization rate; Net $R_n$, net nitrification rate. The bolder frame presented the larger value.

is higher than its production might mean that microorganisms absorb a lot of nutrients to maintain growth and reproduction of their populations (*Miller et al., 2009*; *Liu et al., 2021*), which corresponds to our results of higher soil microbial biomass C and N contents in ErM forest (Figs. 4A and 4B). Moreover, some studies showed that ErM fungi had wider and stronger ability to decompose organic matter than EcM fungi (*Read, 1991*; *Read & Perez-Moreno, 2003*).Therefore, we speculated that ErM forest had higher demand of organic N and soil net N mineralization was dominated by net ammonification.

## CONCLUSIONS

Soil C and N cycling patterns in three primary forests with different mycorrhizal types were compared in our study. AM and EcM forests had relatively faster soil C cycling than that in the ErM forest. Further, the AM forest had lower soil C:N, higher soil inorganic N content and soil net N mineralization rate, indicating AM forest might absorb more soil inorganic N. EcM and ErM forests might demand more organic N sources. These findings provide an in-depth understanding on differences in the way of nutrient acquisition of forests with different mycorrhizal types, which plays an important role in evaluating soil C and N cycling of the forest ecosystems on the eastern Qinghai-Tibetan Plateau.

## ACKNOWLEDGEMENTS

We thank Qiuhong Feng from Sichuan Academy of Forestry for her help during sampling. We are also grateful to the reviewers and academic editor for their professional comments and suggestions.

### Funding

This work was supported by the Scientific Research and Development Project of the Ecology and Nature Conservation Institute, the Chinese Academy of Forestry (CAF) (99805-2020) and the Fundamental Research Funds of CAF (CAFYBB2018ZA003). The funders had no role in study design, data collection and analysis, decision to publish, or preparation of the manuscript.

### Grant Disclosures

The following grant information was disclosed by the authors:
Scientific Research and Development Project of the Ecology and Nature Conservation Institute, Chinese Academy of Forestry (CAF): 99805-2020.
Fundamental Research Funds of CAF: CAFYBB2018ZA003.

### Competing Interests

The authors declare there are no competing interests.

## Author Contributions

- Miaomiao Zhang conceived and designed the experiments, performed the experiments, analyzed the data, prepared figures and/or tables, authored or reviewed drafts of the article, and approved the final draft.
- Shun Liu conceived and designed the experiments, performed the experiments, authored or reviewed drafts of the article, and approved the final draft.
- Miao Chen performed the experiments, prepared figures and/or tables, and approved the final draft.
- Jian Chen performed the experiments, prepared figures and/or tables, and approved the final draft.
- Xiangwen Cao performed the experiments, prepared figures and/or tables, and approved the final draft.
- Gexi Xu conceived and designed the experiments, prepared figures and/or tables, and approved the final draft.
- Hongshuang Xing analyzed the data, prepared figures and/or tables, and approved the final draft.
- Feifan Li analyzed the data, prepared figures and/or tables, and approved the final draft.
- Zuomin Shi conceived and designed the experiments, authored or reviewed drafts of the article, and approved the final draft.

## Field Study Permissions

The following information was supplied relating to field study approvals (*i.e.*, approving body and any reference numbers):

Field experiments were approved by Ecology and Nature Conservation Institute, Chinese Academy of Forestry (project number: 99805-2020).

## Data Availability

The raw data is available in the Supplemental Files.

## Supplemental Information

Supplemental information for this article can be found online at http://dx.doi.org/10.7717/peerj.14028#supplemental-information.

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
