# Peer review of "The below-ground carbon and nitrogen cycling patterns of different mycorrhizal forests on the eastern Qinghai-Tibetan Plateau"

_PeerJ, doi:10.7717/peerj.14028_

## Round 0.1 · original submission · Major Revisions

Dear Dr Miaomiao,

Sorry for the late assessment but it is increasingly difficult to find competent reviewers. Anyway, we received two thorough revisions from experts in the field, which consistently provide a major revision indication for your paper, Nevertheless, both reviewers, appreciate your study and I would also welcome a revised version of your manuscript.

I recommend a point-to-point answer to the reviewers' observations and a revision performed accordingly.

Sincerely,

Leonardo Montagnani

Reviewer 1 ·

Basic reporting

• Although the authors meaning come across in the manuscript the English language and particularly grammar used in needs some work before it is to a professional standard.
• Introduction provides a good background for the context of the paper.
• Figures are well made, however some captions need work and figure 5 is particularly confusing.
• Raw data is supplied, however units are lacking throughout. Please include these to make the raw data interpretable by others.

Experimental design

• Original and primary research, which clearly a lot of effort has been put into.
• Hypotheses are well defined and meaningful.
• The research carried out has clearly taken a lot of work and has been done to a high standard.
• The methods could use more detail in order to be fully reproducible.

Validity of the findings

• All underlying data is provided and seems robust.
• Conclusions drawn are somewhat outwith the scope of the data collected in some cases. Considerable work needs to be put into the discussion section of the paper in order to improve its flow, sequence of arguments and narrative. Please see general comments for further comments on this.

Additional comments

I enjoyed reading this paper in which the authors compare the biogeochemistry and ecosystem function for three forest types defined by dominant mycorrhizal associations. It is clear that a lot of work has gone into collecting these data and they will make a valuable contribution to the field, however, some points made seem somewhat outwith the findings of the data presented. I urge authors to consider what they can state as findings and what is speculation and to be explicit about this. This is by no means to devalue the data presented, but merely be more aware of and clear about its limitations. I have included some further comments below which the authors may wish to consider.

General comments:
1. The abstract reads well and the outline of the study is well presented. However it would be useful to briefly have some more details on the methods used and also some more specific numbers on the trends described in findings part of the abstract.
2. In the introduction you seem to group ErM and EcM fungi together and contrast them with AM fungi, however ErM fungi have distinctive ecosystem effect which differ from EcM fungi. See for example Clemmensen et al. 2021. It may be worth introducing some of this nuance into this section.
3. Please be more specific about techniques used. In the methods section the authors describe creating sub-samples of mineral soils to be analyser for various different characteristics such as DOC, microbial biomass C and N and total N, however no information is given on how these things were measured, which techniques or instruments where used. Please include much more detail in this and other sections of the methods to make it clear to the reader how these data were obtained. Some detail is included in the ‘auxiliary measurements’ section, however this seems like a rather odd place to put this information. It would seem more appropriate to include this when the variables are first mentioned, however that may just be personal preference. If you chose to keep the current formatting I would suggest at least indicating that more details follow in ‘x’ section.
4. It is clear that a lot of work has gone into data collection for this paper and I applaud the authors for that.
5. In the results section it would be helpful for the reader to have some kind of confidence interval for the numbers you report e.g. lignin:N ratio (x ± y) in order to understand the magnitude of variation in text as well as in the figures. This may be a journal specification, in which case please do stick with what the journal require.
6. The discussion section of this paper needs significant work. In its current for it is very hard to follow, partly due to grammatical errors and partly due to the flow of the paragraph. Consider arguments carefully and how they build logically on from one another, at the moment statements seem thrown in with very little context. Maybe consider using subheadings to separate arguments into distinct sections? Lastly, some statements made are rather bold and not necessarily underpinned fully by data presented here. Make sure to be explicit about this.
7. Throughout the manuscript attention needs to be paid to the language and grammar used to ensure that an international audience can clearly understand your text. I suggest you have a colleague who is proficient in English and familiar with the subject matter review your manuscript, or contact a professional editing service.

Specific comments:
L30: please consider changing to: “(ErM) trees respectively were studied on the eastern Qinghai-Tibetan Plateau.”
L47-49: This sentence does not make grammatical sense. Please consider re-writing.
L52-54: What is the mechanism for this? Why does competition with free-living fungi increase soil C? Are you suggesting the Gadgil effect is at play here?
L64: Please check the grammar of the start of this sentence.
L67: This should be “..different types of mycorrhizas…”.
L116-118: Please define what you mean by primary forest. Was there a certain percentage of each species required to be considered primary?
L125: Did the installation of the collars at 5cm depth cut lots of roots in the process and do you think this will have affected your measurements significantly?
L133: You say “It was extracted..” what is “It” in this case? Be explicit about this please.
L133-: Please elaborate on how air samples from your chamber taken a these time intervals was converted into a CO2 flux measurement. The technique used is perfectly valid but please give more details on how you get from this to the data presented.
L135: What do you mean by “transported to the laboratory in time”?
L157: The past tense of grind should be ground.
L198: I do not find presenting the results as x% higher very clear. You could add the actual mean values with confidence intervals or standard deviation here.
L203: As the relationship between soil respiration and temperature is fairly well established I find it somewhat surprising that you found no correlation in your data. Is that because you talk about C cycling as a whole rather than specific processes within that, e.g. soil respiration?
Figure 1: Please check grammar of caption.
Figure 5: I find the use of percentages in this figure confusing and not very intuitive. Is there maybe a better way of presenting these differences? For example you could combine all three mycorrhizal types on one figure but colour the numbers for each one differently, then the reader would be able to compare them easily.
L207: This is a very bold statement, talking about forest carbon cycling based on measured respiration alone. I would advise amending it to reflect the measurements taken with the possibility of speculating on what this might mean for carbon cycling as a whole.
L212: Simply stating “These could support our results (Fig. 3a).” without any further elaboration is not sufficient here. Please expand on this in more detail to take the reader with you in your line of thinking.
L217: Pathways should be pluralised in this sentence.
L227-228: Refer to the data figures for these results rather than the summary figure 5.
L230: Do you mean “use of” here?
L240: Please check the grammar of this sentence.

·

Excellent Review

This review has been rated excellent by staff (in the top 15% of reviews)
EDITOR COMMENT
It is a pleasure to receive comments like this. They are correct informative and help the authors to enhance their scientific level. Unfortunately, the academic career is based only on the publications, and not on the reviews. I would support a change in this direction, to include reviews, if possible. Sincerely, Leonardo Montagnani

Basic reporting

This paper explores the differences in soil C and N cycling between forests dominated by trees that associated with AM, EcM, and ErM fungi in the Qinghau-Tibetan Plateau. This is done using pool and flux measurements of important C and N transformations and pools such as CO2 efflux, net N mineralization, leaf litter chemistry, and microbial biomass. The authors found that AM forests had higher C and N fluxes than EcM and ErM forests. This adds welcomed replication data to the Mycorrhizal Associated Nutrient Economy (MANE; Phillips et al., 2013, New Phytologist). Notably, this paper makes the much needed, yet not often included, distinction between ErM- vs. EcM-dominated forest system. The message of this paper, however, is somewhat obscured by problems with the writing and grammar throughout the manuscript. These issues range from incomplete and unclear sentences to sentences in the discussion where it is uncertain if the results the authors cite are derived from their own work or the work of others. These problems make it difficult to understand the implications of the actual findings of this study. I have included selected examples of how and/or where the language could be improved below.

Verb tenses in the introduction should be consistent. One example of where this could be improved can be found on line 59 where “were relatively slower” should read as “are relatively slower.” This same change could be applied throughout the introduction.
Line 25: “plays” should read as “play”
Line 44: should be “stress resistant”
Line 64: “That is possible” should read “This is possibly attributed to”
Lines 74-75: rewrite first sentence for clarity
Line 85: define what “changing the trait” means
Lines 95-96: “Is abundant in natural resources” should be removed
Line 133: unclear what “it” refers to in the sentence beginning with “It was extracted”
Lines 239-242: rewrite for clarity and matching verb tense to subject
Line 251: Unclear what “ammonification consumption vs. production” means. Please clarify

Overall, this paper does a good job of situating itself within existing research and introducing concepts relevant to the hypothesis. However, there are some problems with contextualization in the introduction. Specifically, key papers and hypotheses are miscited or misconstrued. I have provided one major example and a few others by line numbers.

Given the comparisons between AM, EcM, and ErM, systems, the MANE framework (Phillips et al., 2013, New Phytologist) is central to this paper. However, there are two places where it is miscited. In line 91, the MANE framework is miscited as a global analysis. The paper contains an experiment that was conducted in Indiana, the results of which were synthesized with literature to propose a broader hypothesis. Additionally, in line 78 it is stated that the MANE framework “verified” previous findings, which is incorrect. The MANE framework, rather, unifies observed differences between AM and EM forests. This could be corrected by replacing “verified” with “described.”

Lines 52-54: This sentence concerning the C-implications of inter-guild competition, does not have sufficient support in the literature. There is not yet a clear link between N competition and soil C contents, as most examinations are restricted to the litter layer (but see Sterkenberg et al., 2018, ISME for an analysis of boreal forest organic layer). Additionally, the paper cited in this sentence (Lei et al. 2018, Science), is primarily concerned with AM fungi, not with EcM or ErM as stated. This sentence should be removed, as it is not necessary to the context.
Lines 61-64: The MEMS framework (Cotrufo et al., 2013, Global Change Biology) refers to the stabilization of C on minerals in different systems, not the quantity of C and N, as is implied in this sentence. Direct experimental evidence should be cited here.
Lines 103-105: It would be useful to outline the specific metrics defining faster and slower N cycling as well as inorganic and organic nutrient economies, as these terms may be misunderstood by a broad audience.

The presentation of the results of the study should be elaborated upon and, in some cases, lack clear displays of statistical support. A major area for improvement for the presentation of the results is to include the numerical values of the pools/fluxes for the different forests rather than (or at least in addition to) the percent differences between the forest types. This will not only allow for easier comparisons between different pools and fluxes in this paper, it will also allow for more detailed comparisons with other published work. Examples of where this change could be implemented can be found in lines 182-184, however it should be applied to the entire section. Additionally, in all cases where statistical significance is mentioned a p-value (or other relevant summary statistic) should be included after the result.

The raw data are included, however they lack metadata or units for the reported variables. While these are provided in the body of the text, they should also be included in the raw data sheet. Overall the figures were clear and well labelled. The only exception to this is figure 5, which outlines the percent differences in pools and fluxes. To improve this figure, I would recommend that instead of presenting the two-way percent differences, the values in the figures should be the means for that pool and flux in that forest. For ease of comparison, the boxes and arrows could have their size scaled to the value (e.g. thicker arrow denotes larger flux).

Experimental design

The gap in knowledge and hypotheses that this study address are clearly stated and are important to the field. This is especially true of the inclusion of ErM fungi as separate from EcM fungi, which is not addressed in other, similar studies.

While the methods used to address these questions are appropriate, they are not always sufficiently described in the Materials and Methods section. There are two major ways in which this section could be improved, followed by minor points.

First, while the description of the plant community of this site is thorough, other aspects of the site description could be improved. Given the centrality of soil to this study, the manuscript would be improved by describing the soil texture, taxonomy, and other relevant edaphic factors (pH, parent material, etc). Also, it should be noted how far apart the different forests are. Second, there are certain chemical assays, which are either not described (for an example, see the determination of lignin, line 157) or are described below where they are first mentioned. One way to address the latter issue would be to remove the “Auxiliary measurements” subsection and move that text to the end of the “Soil sampling and N mineralization” subsection.

Apart from these suggestions relating to presentation of the methods, I also feel as though the PVC core method should be elaborated upon. Specifically, the authors state (line 125) that the PVC cores used to measure soil respiration were buried 5 cm into the soil. This design severs mycelial connections to the measured area, leading to reduced mycorrhizal influence on respiration. While some longer-exploration type mycorrhizal fungi may grow back in, this is not possible for AM fungi, which tend to produce less extensive hyphal networks. Given the interpretations applied to the soil C flux, this manuscript would be improved by either an explanation of this limitation, or by framing the CO2 flux primarily as heterotrophic respiration, and as a metric of SOM decomposition.

I have included further minor examples of where this section could be improved by line number below.

Line 132: The authors need to define what “weather was good” means. What metrics were used to determine when to measure?
Line 141-145: Unclear if these PVC collars are the same as those used to study respiration, if not, their purpose/use should be more clearly defined.
Line 154: “Good growth trees” should be defined. What metrics were used to determine which trees were sampled
Line 157: The cited lignin quantification method should be briefly described
Line 220: The idea of using microbial biomass C:N ratio as a proxy for fungi:bacteria ratios should be introduced in the Materials and Methods section

Validity of the findings

While the conclusions made in the discussions and conclusion sections of this article are well-linked to the existing literature, the links with the stated results are unclear. One major issue in this regard, possibly related to the problems with language discussed above, is that it is difficult to distinguish what the authors are concluding based on the findings of this specific study from where the authors are drawing on supporting studies. I have included some examples (by line) of instances of this below. I recommend that these examples, and the discussion section, generally, are rewritten to make clear what is being concluded from data in this study, as opposed to data from the published literature. An example of this being done well in the present manuscript is the final paragraph of the discussion (lines 258-271). This may be a useful reference in rewriting.

Lines 208-209: This sentence, as written, seems to imply that organic N degradation was measured and assessed in this study, but was not.
Line 214: The contribution of plant-derived C to the soil was not measured in this study.
Lines 235-237: Mineral associated organic matter was not measured in this study.

---

## Round 0.2 · Major Revisions

Dear Dr Shi,

I evaluated personally the revised version of your paper. I believe it can be published after you have checked, and eventually corrected, the following points. Please provide detailed answers to the points I raised and a revised version of your manuscript.

I see still a major problem: you consider that, based on the statistical analysis performed, there are no significant relationships between environmental variables and the flux observed in the three forest typologies. However, the indication about the meteorological setup is poor, and the few indications provided suggest that it was possibly inadequate. I recommend removing any indications related to this subject, apart from recommending the need for a better characterization of the soil meteorological variables, possibly influencing plant-fungi-bacteria interactions.

Specific comments:

Abstract, line (L) 27: ‘… mycorrhizal types are still lacking, I would say that it is still incomplete.

L 33: Please explain what you mean by 'lower quality of the litter.

L52: represent->have

LL68-72: please reword, I cannot understand.

L121: ‘The primary forest present climax community…’: if you do not provide evidence about the supposed lack of disturbances, please skip the sentence.

L138: ‘temperature probe’. Which model? Which sensor accuracy and precision? Measured during which period? Data memorized in which way? If all this is missing, better remove this section.

L148: What was the model and make of the static chamber used? All the experiment details must be fully reproducible.

L163: What do you mean with ‘dried naturally’? at room temperature?

L231: ‘pathway’, pathways instead (?)

L261: the term ‘refractory is generally used in marine studies. Recalcitrant is more common?

Op-> Op de Beeck

L278: Please consider removing the comment about the temperature effect since the experimental set-up was possibly inadequate. The same on lines 295-296.

Figure 1: On the X axis, the values report a fraction or a percentage.

Figure 2: Kg or kg?

The figure 5 legend appears incomplete

In Table 1, could you report the geographical coordinates and the elevation of the three forests?
Could you better explain what you mean by ‘canopy density’? In which units? Measured in which way?
It is not clear at all what are the input data in Tables 2 and 3. Consider removing or providing all the details.

Sincerely,

Leonardo Montagnani

---

## Round 0.3 · accepted · Accept

Dear Dr. Miaomiao,

I am pleased to inform you that I consider your paper acceptable now.

Congratulations!

Sincerely,

Leonardo Montagnani